# Deep Credal Neural Network: Characterization of Imprecision Between Categories

## Abstract

Quantification and reduction of uncertainty in deep learning techniques have received much attention but ignored how to characterize the imprecision caused by such uncertainty. In some tasks, we prefer to obtain an imprecise result rather than being willing or unable to bear the cost of an error. For this purpose, we present a deep credal neural network (DCNN) based on the theory of belief functions, aiming to assign samples that are indistinguishable for specific categories to the union of these, called meta-category. In DCNN, a designed mechanism assigns multiple labels to some training samples to constrain the known loss functions. Once assigned, it indicates that these samples may be in an overlapping region of different categories, or the original label is wrong. Afterward, the training labels are reconstructed and therefore classify the test samples. Once assigned to meta-category, the prediction of this test sample is imprecise. Experiments based on some remarkable networks have shown that DCNN can not only improve accuracy but also reasonably characterize imprecision both in the training and test sets.

## 1   Introduction

Deep neural networks have achieved remarkable success in a wide range of computer vision tasks (Le Cun et al. [2015]), including image classification (Deng et al. [2009]), and are still moving toward greater speed and accuracy (Szegedy et al. [2015], Girshick [2015], Ren et al. [2015]), However, imperfect knowledge (data uncertainty) (Gal and Ghahramani [2016], Hüllermeier and Waegeman [2021]) runs counter to our desire to train perfect prediction networks, and for this reason, many new approaches (Abdar et al. [2020]) have been proposed focusing on quantifying and reducing data uncertainty. In fact, if data uncertainty inevitably arises, then focusing on the imprecision caused by them will have a significant positive impact on the training and test sets[1]. Let's take some realistic images as an example, as shown in Fig. 1.

For image classification tasks, these data are imperfect. We can find that data uncertainty may be caused by many factors such as shooting angles (Fig. (a)) and occlusions (Fig. (d)) that make some different species look similar (Figs. (a), (b)) or different species appear in one image (Fig. (c)), and even some are mislabeled. At this point, the network is not only unable to extract the distinctive features of the category from these images but also restricted. In this case, it may be a better choice to filter these images from the original category. Besides, it is also difficult to classify these images precisely in the test set because they do not have the distinctive features of a category. In fact, we

---

[1]In this work, uncertainty refers to the lack of certainty, a state of limited (insufficient) empirical information (knowledge) where it is impossible to exactly describe the state. In contrast, imprecision refers to the lack of precision, a state of fuzzy (imprecise) empirical information (knowledge). For example, *"we are not sure it will rain tomorrow"* is uncertain information, and *"it is raining a lot"* is imprecise information because we don't know exactly how much water.

Submitted to 35th Conference on Neural Information Processing Systems (NeurIPS 2021). Do not distribute.

prefer to obtain an imprecise result in some cases rather than being willing or unable to bear the cost of an error. Thus, in this work, we focus on the characterization of imprecision in the training and test sets caused by data uncertainty.

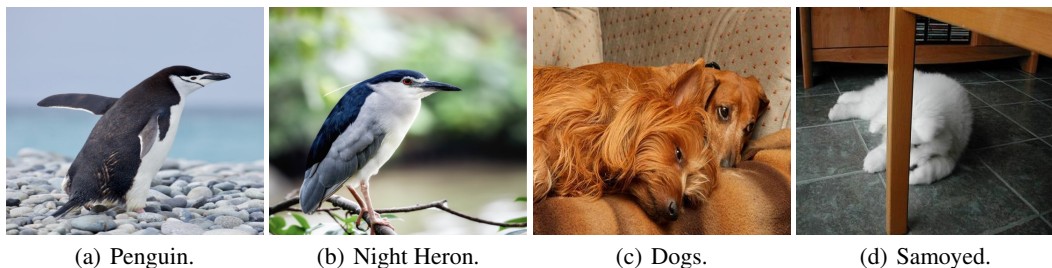

(a) Penguin.  (b) Night Heron.  (c) Dogs.  (d) Samoyed.

Figure 1: Illustration of uncertainty and imprecision.

Theory of Belief Functions (TBF) introduced by Shafer in his Mathematical Theory of Evidence (Shafer [1976]), also known as Dempster–Shafer Theory, is appealing for dealing with such uncertain and imprecise information (Shafer [1976], Denœux [2019a]). In TBF, a mass function $m$ is used to characterize uncertainty, and meta-category, defined as the union of different specific categories, characterizes imprecision. Once assigned to meta-category, for example, one image in Fig. 1, it indicates that the category of this image is imprecise (Zhang et al. [2021]). In other words, if an image has two or more potential category labels, we call it an imprecise image. A few works (Denœux [2000], George and Sankaran [2019], Tong et al. [2021]) combining deep neural networks and TBF have been proposed. In essence, they only modify the output of the network. By doing this, they can characterize the imprecision of the test set but not the training set. In addition, they ignore the impact of the imprecision on the network, making it difficult to improve accuracy.

In this work, we propose a TBF-based deep credal neural network (DCNN) to characterize the imprecision in the training and test sets and thereby constrain the network training to improve the classification performance. In DCNN, we first design a label assignment mechanism, aiming to assign multiple labels to some training samples and thus constrain training based on this. When the network reaches the global optimum, some training samples will be labeled belonging to multiple potential categories to characterize the imprecision in the training set. Then, we reconstruct the training set, which consists of precise samples with specific category label and imprecise samples with multiple category labels. This further extraction of prior knowledge is eventually used to retrain and improve the network and thereby classify the test set. Since we can extract some common features of different categories from imprecise training samples, this can provide a good guide for classifying test samples that share similar features, thus avoiding the risk of being misclassified.

We choose seven remarkable networks (VGGNet16 (Simonyan and Zisserman [2014]), Resnet101 (He et al. [2016]), GoogLeNet (Szegedy et al. [2015]), MobileNetV2 (Sandler et al. [2018]), DenseNet169 (Huang et al. [2017]), EfficientNetB0 (Tan and Le [2019]), ShuffleNetV2 (Ma et al. [2018])) to evaluate the DCNN on two image classification benchmarks (Imagewoof-5, and Flowers). The results show that DCNN can reduce the error rate and improve the accuracy while reasonably characterizing the imprecision of both in the training and test sets.

## 2 Basics of belief functions

We only introduce some basic notions of the Theory of Belief Functions (TBF) used in this work. For a $n$ classification task with the frame of discernment $\Omega = \{c_1, ..., c_n\}$, TBF extends it to the power-set $2^\Omega$, which contains all subsets of $\Omega$. For example, if $n = 3$ then $\Omega = \{c_1, c_2, c_3\}$ and we have $2^\Omega = \{\emptyset, c_1, c_2, c_3, \{c_1, c_2\}, \{c_1, c_3\}, \{c_2, c_3\}, \Omega\}$. The meta-category $\{c_i, c_j\}$, considered as a new category and defined as the union of singleton (specific) categories $c_i$ and $c_j$, represents the possibility of an sample belongs to either $c_i$ or $c_j$. Once assigned to meta-category, we can say that the sample is imprecise. In other words, imprecise samples usually have common features of different categories, and they are at high risk of misclassification once they are forced to be assigned to a singleton (specific) category. In contrast, we can say a precise sample if it belongs to one singleton (specific) category (Zhang et al. [2021]). A mass function $m$ between 0 and 1 is given to each

subset of $\Omega$, i.e. $m : 2^\Omega \to [0, 1]$, whenever it verifies two axioms (Denœux [2008]): $m(\emptyset) = 0$ and $\sum_{A \subseteq 2^\Omega} m(A) = 1$. The category $A$ can be either singleton with $|A| = 1$ or meta-category with $|A| \geq 2$, where $|A|$ is the number of singletons included in the category $A$. Also, $m(A) \in [0, 1]$ represents uncertain degree of the sample belonging to the category $A$.

## 3 Deep credal neural network

In classification tasks, we hope to use training samples that significantly represent different categories to train the network and thereby extract distinctive features of each category. In this case, identifying those imprecise (fuzzy) or mislabeled samples in advance helps to train a higher performance network. In addition, the features of these training samples are often similar to that of imprecise samples that are prone to be misclassified in the test task, thus extracting the features of these training samples likewise contributes to characterize the imprecision of the test ones. The contributions of DCNN can be summarized in two parts: 1) Reassign labels for training samples, and 2) Reconstruct training categories (labels) and classify test samples. The overall of the DCNN is presented in Fig. 2.

### 3.1 Reassign labels for training samples

The purpose of this subsection is to retain one and only one label for training samples with distinctive category features while assigning potential labels to those training samples that are imprecise or mislabeled. In most cases, samples are indistinguishable between at most two categories. Thus, we simplify to allow at most one additional potential label (category) to be assigned to one sample. Assume that $x$ represents a training sample belonging to the category $c_{ture}$, and the corresponding label is encoded with one-hot. The position corresponding to $c_{ture}$ in an $n$-dimensional all-0 matrix is set to 1, denoted as $\mathbf{y} = [\phi(c_1), ..., \phi(c_n)]$ subject to $\phi(c_i) = \begin{cases} 1, c_i = c_{ture} \\ 0, c_i \neq c_{ture} \end{cases}$, $i = 1, ..., n$. During training, the predicted result is represented by $\hat{\mathbf{y}} = [p(c_1), ..., p(c_n)]$, where $p(c_i)$ denotes the predicted probability of being assigned to $i$-th category. To facilitate the assignment of potential labels, the following definitions are provided: $c_{ture}$ is the original given category of $x$, $c_{\max}$ is the category with the highest predicted probability, and $c_{\max 2}$ is the category with the second-highest predicted probability, where $c_{ture}, c_{\max}, c_{\max 2} \in \Omega$. During training, whether assign an additional potential label (category) to $x$ is based on two judgment criteria.

- When category $c_{\max}$ is not $c_{ture}$ in the prediction result, $c_{\max}$ is considered as a potential label (category). In this case, we have $\delta_1 = p(c_{\max}) - p(c_{ture})$, $\delta_1 \in [0, 1]$. If $\delta_1 \to 1$, we consider that $c_{ture}$ is mislabeled. In contrast, if $\delta_1 \to 0$, we consider that $x$ may have common features of both categories $c_{\max}$ and $c_{ture}$, or not have distinctive features of one of these two singleton categories.

- When category $c_{\max}$ is $c_{ture}$ in the predicted result while the predicted probabilities of $c_{\max}$ ($c_{ture}$) and $c_{\max 2}$ are very close, we likewise consider that $x$ may have common features of both categories $c_{\max}$ ($c_{ture}$) and $c_{\max 2}$, or not have distinctive features of one of these two singleton categories. In this case, we have $\delta_2 = p(c_{\max}) - p(c_{\max 2})$, $\delta_2 \in [0, \alpha]$, and $c_{\max 2}$ is considered as the potential category[2]. In contrast, the larger $\delta_2$ is the less necessary to assign the potential label $c_{\max 2}$.

Based on the above analysis, it is assumed that the potential label assigned is $\dot{\mathbf{y}}$, encoded in the same way as $\mathbf{y}$ and denoted as $\dot{\mathbf{y}} = [\varphi(c_1), ..., \varphi(c_n)]$, $\forall i, \varphi(c_i) = 0$. A label assignment mechanism is designed and defined by:

$$\dot{\mathbf{y}} = \begin{cases} \varphi(c_{\max}) = 1, & \text{if } c_{\max} \neq c_{ture} \\ \varphi(c_{\max 2}) = 1, & \text{if } c_{\max} = c_{ture}, \delta_2 \leq \alpha \\ \varphi(c_{ture}) = 1, & \text{if } c_{\max} = c_{ture}, \delta_2 > \alpha \end{cases} \tag{1}$$

From Eq. (1), we can find that some imprecise samples are obtained for each epoch. This imprecision in the training set will constrain the network by our redefined loss function during training. In this step, the chosen network is exploited without modifying its structure, and the weights and bias term

---

[2]$\alpha$ is a given parameter controlling the number of imprecise samples in this case. $\alpha = 0.01$ is the default in this work, and we will discuss parameter $\alpha$ later.

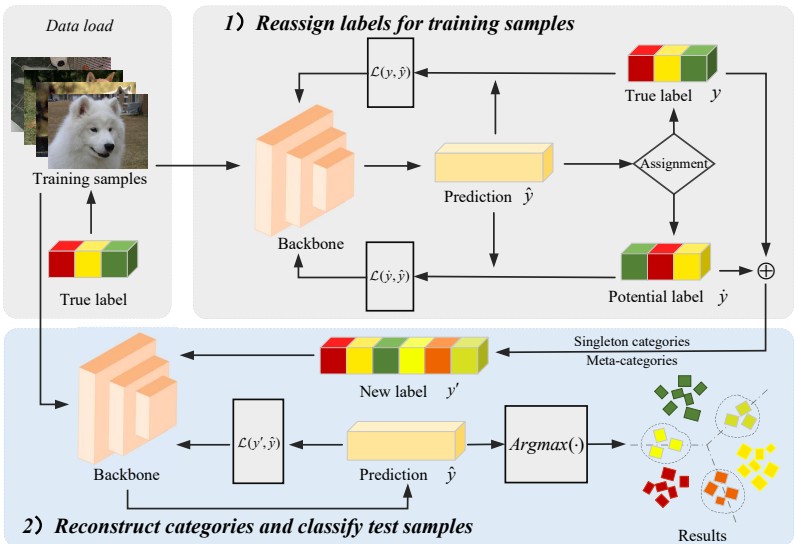

Figure 2: The proposed deep credal neural network (DCNN).

parameters are initialized randomly. Considering that the true label $\mathbf{y}$ and the potential label $\dot{\mathbf{y}}$ have different effects on the network in different epochs, we set a weighting factor $S(t)$ to balance the effects of these two labels on the loss function. The involvement of one more label makes the decision more reliable and can reach a smaller overall loss. We hope $\mathbf{y}$ to dominate the role at the beginning of the training, and the role of $\dot{\mathbf{y}}$ on the loss function gradually strengthened. Thus, a function that gradually rises from 0 to a certain upper limit is needed. Based on Sigmoid function (Cybenko [1989]), we define the new weighting factor $S(t)$ as follows:

$$S(t) = \frac{1}{1 + e^{-\frac{3}{\beta-1}(t-1)}} - \frac{1}{2}, \ t = 1, 2, ..., epoch \tag{2}$$

where $t$ is the training epoch. We have $S(\beta) = \sigma(3) - 0.5 \approx 0.4526$ if the training epoch $t$ reaches $\beta$. Here $\beta$ is a given parameter, and $\beta = 20$ is the default. These two functions are shown in Fig. 3.

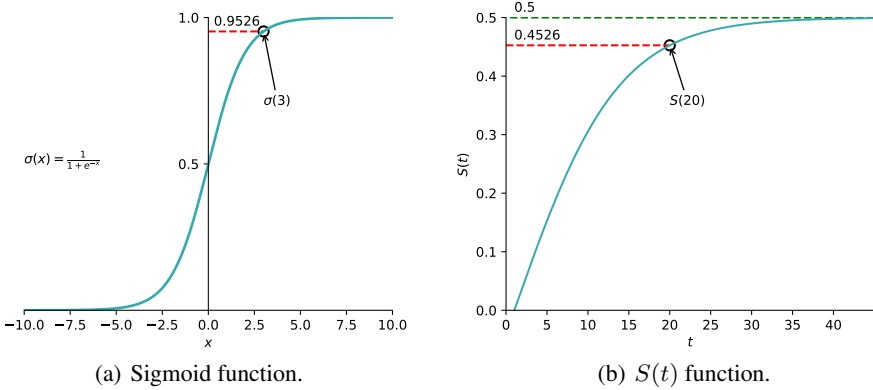

(a) Sigmoid function.  (b) $S(t)$ function.

Figure 3: Illustration of Sigmoid function and redefined $S(t)$ weighting factor.

The loss functions are defined with $\delta_1$ and $\delta_2$ during training, and the larger their values the greater the penalty to training, *i.e.* the greater their effect on the loss function. Considering that $\delta_1 \in [0, 1]$ and $\delta_2 \in [0, \alpha]$, we normalize $\delta_2$ and use $\zeta = \frac{\delta_2}{\alpha} \in [0, 1]$. Taking the multi-classification cross-entropy

loss function as the basis, the loss function $L(\mathbf{y}, \dot{\mathbf{y}}, \hat{\mathbf{y}})$ is redefined by:

$$\mathcal{L}(\mathbf{y}, \dot{\mathbf{y}}, \hat{\mathbf{y}}) = \begin{cases} [-\sum\limits_{i=1}^{n} \mathrm{y}_i \log \hat{\mathrm{y}}_i] - S(t) \cdot e^{-\delta_1} \cdot [-\sum\limits_{i=1}^{n} \dot{\mathrm{y}}_i \log \hat{\mathrm{y}}_i], & \text{if } c_{\max} \neq c_{ture} \\ [-\sum\limits_{i=1}^{n} \mathrm{y}_i \log \hat{\mathrm{y}}_i] - S(t) \cdot e^{-\zeta} \cdot [-\sum\limits_{i=1}^{n} \dot{\mathrm{y}}_i \log \hat{\mathrm{y}}_i], & \text{if } c_{\max} = c_{ture}, \ \delta_2 \leq \alpha \\ [-\sum\limits_{i=1}^{n} \mathrm{y}_i \log \hat{\mathrm{y}}_i], & \text{if } c_{\max} = c_{ture}, \ \delta_2 > \alpha \end{cases} \quad (3)$$

where $e^{-\delta_1}, e^{-\zeta} \in [\frac{1}{e}, 1]$. The label assignment mechanism is executed once per epoch during training, and the assigned label obtained from the last epoch is the final result when the network training stabilizes. By doing so, we can obtain a small number of imprecise training samples to characterize the imprecision of the training set. In fact, removing these imprecise samples from the original singleton categories help the network to extract distinctive features of different categories from the remaining training samples.

### 3.2 Reconstruct categories and classify test samples

When the label assignment is complete, only a small number of training samples have two labels in most cases. At this point, the labels of these imprecise samples can be checked artificially and corrected if a labeling error can be identified. This is much easier than sifting through the original dataset to find the wrong labels. In contrast, if the imprecision of these samples is caused by their own uncertainty, then we continue to keep the two labels. These imprecise samples cannot be retained in the original category because they contain features from multiple categories, which would reduce the network's ability to recognize that category. In fact, it is a good opportunity to use these imprecise samples to characterize some common features between different categories. From the view of data distribution, these samples are likely to be precisely distributed in the overlapping regions of the different singleton categories. In this case, the test samples with features similar to these imprecise training samples are at risk of being misclassified if forced into one singleton category.

Based on TBF, these imprecise training samples can be considered as the new training samples in meta-categories which are considered as new categories. We can extract the distinctive features of the meta-categories based on these imprecise samples. Thus, the training samples are reconstructed as a dataset containing $\frac{n(n+1)}{2}$ categories[3]. In this case, we can redefine a new frame of discernment as $\mathcal{M}^{\Omega} = \{c_1, ..., c_n, c_{1,2}, ...., c_{n-1,n}\}$. For the training sample $x$, its new label can be encoded as $\mathbf{y}' = [\psi(c_1), ..., \psi(c_n), \psi(c_{n+1}), ..., \psi(c_{\frac{n(n+1)}{2}})]$ based on $\mathbf{y}$ and $\dot{\mathbf{y}}$. For example, if $\mathbf{y} = [1, 0, 0]$ and $\dot{\mathbf{y}} = [0, 1, 0]$, we have $\mathbf{y}' = [0, 0, 0, 1, 0, 0]$ which is considered as the new label for $x$. After we reconstruct the training categories and the corresponding training samples, we retrain the network using the multi-classification cross-entropy loss function, defined by:

$$\mathcal{L}(\mathbf{y}', \hat{\mathbf{y}}) = - \sum_{i=1}^{\frac{n(n+1)}{2}} \mathrm{y}'_i \log \hat{\mathrm{y}}_i \quad (4)$$

The trained network can be used to classify the test samples. Since the new framework contains meta-categories, the output for each test sample can be regarded as a mass function with $\sum\limits_{i=1}^{\frac{n(n+1)}{2}} m(A_i) = 1$ and used for the final decision-making. Once a test sample is assigned to a meta-category $A_i$ with $|A_i| = 2$, and subject to:

$$m(A_i) = \max\{m(A_1), ..., m(A_{\frac{n(n+1)}{2}})\} \quad (5)$$

it indicates that the test sample does not have the distinctive features of one of the two singleton category included in $A_i$. Assignment to meta-category is an imprecise result, but it also reduces the risk of error. In some applications, this cautious decision-making is very important. In this case, other techniques can be further employed to distinguish these imprecise samples. Although this may increase expenses, we may not be able to bear the cost of an error.

---

[3]It contains $n$ singleton categories and $\frac{n(n-1)}{2}$ meta-categories.

# 4 Experiments

## 4.1 Datasets, indexes, and implementation details

We conduct experiments on two image classification datasets. The detailed statistics such as category numbers and data splits are summarized as follows.

**Imagewoof-5 dataset** is a subset of 5 categories from ImageNet (Deng et al. [2009]) that aren't so easy to classify since they are all dog breeds. They are Australian terrier, Samoyed, Shih-Tzu, Rhodesian ridgeback, and Dingo. Imagewoof-5 consists of 4,687 training images and 2,063 validation images. We randomly split these validation images into the validation and test sets according to 1:1. Since Imagewoof-5 has different sizes, we resize these to $224 \times 224$ before inputting the network.

**Flowers Recognition dataset** consists of 4242 flower images divided into five categories: Chamomile, Tulip, Rose, Sunflower, and Dandelion. There are about 800 images for each category with a low resolution of about $320 \times 240$ and we resize the images to $128 \times 128$ before inputting the network. We randomly split these images into the training, validation, and test sets according to 3:1:1. This dataset is available at *https://www.kaggle.com/alxmamaev/flowers-recognition.*

**Performance indexes.** Due to the introduction of meta-categories, the traditional indexes such as precision ($PE$), recall ($RE$), and f1-measure ($F1$) (Yang [1999]) cannot be used directly in the statistical results, but fortunately, they have been included in the TBF and correspond to evidential precision ($EP$), evidential recall ($ER$), and evidential F1 ($EF1$) (Zhou et al. [2015]). In addition, the error rate ($R_e$), imprecision rate ($R_i$) (Zhang et al. [2021]), accuracy ($R_a$), and benefit value ($B_T$) (Liu et al. [2017]) are also used as performance indexes, where $R_i$ is the proportion of test samples that initially belong to singleton categories but are assigned to meta-categories containing these singleton categories. $B_T$ is a trade-off between $R_e$ and $R_i$. For a test sample, it scores 1 point if it is classified correctly and 0 point if misclassified, and $(\frac{1}{|A|})^\gamma$ point if assigned to meta-category, where $\gamma = 0.8$ is the default. When there is no meta-category in the results, $B_T = R_a$ and the other indexes degenerate to their counterparts in the probability framework. In summary, the higher the value of these indexes, except for $R_i$, the better. $R_i$ is neutral, which can be adjusted according to what is acceptable to the user.

**Training details.** We conduct the experiments with Pytorch deep learning library. For both datasets, we use a batch size of 32 as the default and reduce it when the model can not fit into the memory. All DCNN frameworks are optimized by using Adam on a single NVIDIA RTX3090 GPU, and the learning rate starts at $10^{-3}$ (only ShuffleNetV2 with $10^{-2}$). We train the network for 30 epochs and decay the learning rate multiply by 0.1 every 20 epochs. Furthermore, the experiments use the EarlyStopping method (Prechelt [1998]) to prevent overfitting. Since DCNN only executes the label assignment mechanism during each epoch, it is consistent with the complexity of the chosen network.

## 4.2 Comparison to remarkable networks.

**The Chosen networks.** We choose 7 remarkable networks to validate the effectiveness of the proposed DCNN, and they are VGGNet16 (Simonyan and Zisserman [2014]), Resnet101 (He et al. [2016]), GoogLeNet (Szegedy et al. [2015]), MobileNetV2 (Sandler et al. [2018]), DenseNet169 (Huang et al. [2017]), EfficientNetB0 (Tan and Le [2019]), and ShuffleNetV2 (Ma et al. [2018]), respectively.

**Our proposed DCNN.** To simulate the case of mislabeling, we randomly labeled 1% of the training images as any other incorrect category. We set up two modes: 1) **DCNN-1**. In this mode, we do not do any processing and directly use the newly reconstructed training set to train the chosen network and then classify the test set. 2) **DCNN-2**. In this mode, after obtaining the newly reconstructed training set, we manually check the imprecise training images and revise these imprecise images that are apparently mislabeled to correctly precise ones. Then, we use the corrected training set to train the network and classify the test set. We record the classification results of these two models separately, and they are the average of 3-5 executions.

**Results. i) Overall.** Tables 1 and 2 show the classification results of the 7 chosen networks and the corresponding DCNNs on both Imagewoof-5 and Flowers Recognition datasets. Specifically,

we studied 7 performance indexes for each network and the corresponding two DCNN models, *i.e.* DCNN-1 and DCNN-2. In both tables, we have highlighted the first two results for each index and highlighted the best result with an underscore. Overall, both models of DCNN outperform the chosen networks in most cases on 6 indexes (except the imprecision rate $R_i$) because of its ability to characterize well the imprecision between different categories in the training and test sets and its ability to extract imprecise images.

Table 1: The results of different networks on Imagewoof-5 dataset

| Methods | $R_e$ | $R_i$ | $R_a$ | $EP$ | $ER$ | $EF1$ | $B_T$ |
|---|---|---|---|---|---|---|---|
| VGGNet16 | 0.3986 | / | 0.6014 | 0.6169 | 0.5982 | 0.5993 | 0.6014 |
| DCNN-1 | **0.3531** | **0.0213** | **0.6256** | **0.6426** | **0.6256** | **0.6330** | **0.6379** |
| DCNN-2 | **0.3443** | **0.0029** | **0.6528** | **0.6551** | **0.6522** | **0.6525** | **0.6544** |
| Resnet101 | 0.3637 | / | **0.6363** | 0.6376 | 0.6359 | 0.6348 | 0.6363 |
| DCNN-1 | **0.3453** | **0.0184** | **0.6363** | **0.6555** | **0.6366** | **0.6446** | **0.6469** |
| DCNN-2 | **0.3511** | **0.0019** | **0.6470** | **0.6572** | **0.6490** | **0.6494** | **0.6481** |
| GoogLeNet | 0.1959 | / | **0.8041** | 0.8053 | **0.8050** | 0.8038 | 0.8041 |
| DCNN-1 | **0.1387** | **0.0863** | 0.7750 | **0.8719** | 0.7752 | **0.8203** | **0.8246** |
| DCNN-2 | **0.1688** | **0.0107** | **0.8205** | **0.8338** | **0.8207** | **0.8269** | **0.8267** |
| MobileNetV2 | 0.3453 | / | **0.6547** | 0.6606 | **0.6563** | 0.6565 | 0.6547 |
| DCNN-1 | **0.2978** | **0.0563** | 0.6459 | **0.7002** | 0.6482 | **0.6711** | **0.6783** |
| DCNN-2 | **0.3152** | **0.0048** | **0.6800** | **0.6881** | **0.6804** | **0.6831** | **0.6827** |
| DenseNet169 | 0.2454 | / | **0.7546** | 0.7542 | **0.7554** | 0.7536 | 0.7546 |
| DCNN-1 | **0.2308** | **0.0155** | 0.7537 | **0.7704** | 0.7552 | **0.7609** | **0.7626** |
| DCNN-2 | **0.1891** | **0.0107** | **0.8002** | **0.8104** | **0.8009** | **0.8052** | **0.8063** |
| EfficientNetB0 | 0.3220 | / | **0.6780** | 0.6847 | **0.6795** | 0.6786 | 0.6780 |
| DCNN-1 | **0.2755** | **0.0650** | 0.6595 | **0.7275** | 0.6607 | **0.6911** | **0.6969** |
| DCNN-2 | **0.3055** | **0.0029** | **0.6916** | **0.6963** | **0.6937** | **0.6925** | **0.6932** |
| ShuffleNetV2 | 0.3104 | / | **0.6896** | 0.6962 | **0.6911** | 0.6913 | 0.6896 |
| DCNN-1 | **0.2949** | **0.0233** | 0.6818 | **0.7108** | 0.6833 | **0.6939** | **0.6952** |
| DCNN-2 | **0.2958** | **0.0029** | **0.7013** | **0.7075** | **0.7022** | **0.7033** | **0.7029** |

**ii) Accuracy and error rate.** Since data uncertainty is inevitable, characterizing the imprecision caused by that uncertainty is a good choice. We find that most mislabeled training images can be extracted and relabeled as precise. In contrast, those images that do not have distinctive category features for various reasons can also be relabeled as imprecise. These imprecise images are assigned to meta-categories to prevent the risk of errors. As a result, the error rate $R_e$ of DCNN is much smaller than that of the chosen network in most cases. For example, our MobileNetV2-based DCNN can reduce $R_e$ on the Flowers Recognition dataset by up to 6% while improving the accuracy ($R_a$) by about 6% at the same time. Thus, our DCNN can not only characterize the imprecision caused by uncertainty but also use this imprecision to constrain the training and thereby improve the classification performance.

**iii) Imprecision rate and other indexes.** Similarly, other performance indexes are definitely better than the chosen network in most cases. However, if more and more images are assigned to meta-categories, $R_a$ decreases. For example, when we use the GooLeNet-based DCNN-1 model on the Imagewoof-5 dataset, it reduces $R_e$ by about 6%, but $R_a$ is about 3% lower than that of GooLeNet, while the imprecision rate ($R_i$) is currently over 8%. Although $R_a$ and $ER$ of GooLeNet are higher than that of our DCNN-1 at this time, this does not mean that the performance of DCNN-1 is lower than that of GooLeNet. In this case, we can find that DCNN-1 outperforms GooLeNet in terms of benefit value $B_T$, i.e. the trade-off between $R_e$ and $R_i$. We can find that if DCNN-2 is chosen, it outperforms GooLeNet in all indexes when $R_i$ is reduced. This again demonstrates that DCNN can not only characterize imprecision but also effectively improve the classification performance. In DCNN, $R_i$ can be controlled by parameter $\alpha$. We can find that the larger the imprecise training images are, the higher $R_i$ of the classification result will be. $\alpha$ can be set manually according to the acceptable imprecision rate in applications.

**iv) DCNN-1 *vs.* DCNN-2.** We can find that the classification performance of DCNN is different in these two modes. In general, the error rate of DCNN-1 is slightly lower or roughly equal to that of DCNN-2, because there are more imprecise training images in DCNN-1 than DCNN-2, which means a larger range of features are extracted for meta-category. However, the low error rate also implies a high imprecision rate, and we can find that the imprecision rate of DCNN-2 is much lower than that of DCNN-1. In addition, the high performance of DCNN-2 is associated with the manual screening of imprecise training images. This indicates that DCNN does have the ability to characterize imprecision caused by mislabels in the training set. In this case, DCNN-1 is suitable for scenarios requiring high execution efficiency but relatively low accuracy. In contrast, DCNN-2 is more suitable for applications requiring high accuracy but relatively low execution efficiency because further screening of imprecise training images may be more costly.

Table 2: The results of different networks on Flowers Recognition dataset

| Methods | $R_e$ | $R_i$ | $R_a$ | $EP$ | $ER$ | $EF1$ | $B_T$ |
|---|---|---|---|---|---|---|---|
| VGGNet16 | 0.3043 | / | 0.6957 | 0.6955 | 0.6970 | 0.6953 | 0.6957 |
| DCNN-1 | **0.2544** | **0.0256** | **0.7200** | **0.7425** | **0.7197** | **0.7301** | **0.7348** |
| DCNN-2 | **0.2555** | **0.0012** | **0.7433** | **0.7437** | **0.7489** | **0.7445** | **0.7440** |
| Resnet101 | 0.3508 | / | **0.6492** | 0.6497 | 0.6478 | 0.6476 | 0.6492 |
| DCNN-1 | **0.3322** | **0.0186** | **0.6492** | **0.6682** | **0.6490** | **0.6562** | **0.6599** |
| DCNN-2 | **0.3287** | **0.0081** | **0.6632** | **0.6702** | **0.6629** | **0.6657** | **0.6679** |
| GoogLeNet | 0.2346 | / | 0.7654 | 0.7688 | 0.7630 | 0.7653 | 0.7654 |
| DCNN-1 | **0.1974** | **0.0209** | **0.7817** | **0.8074** | **0.7787** | **0.7915** | **0.7937** |
| DCNN-2 | **0.2033** | **0.0012** | **0.7955** | **0.7973** | **0.7947** | **0.7958** | **0.7963** |
| MobileNetV2 | 0.3926 | / | 0.6074 | 0.6075 | 0.6052 | 0.6060 | 0.6074 |
| DCNN-1 | **0.3345** | **0.0453** | **0.6202** | **0.6617** | **0.6157** | **0.6340** | **0.6462** |
| DCNN-2 | **0.3310** | **0.0023** | **0.6667** | **0.6746** | **0.6613** | **0.6646** | **0.6680** |
| DenseNet169 | 0.2741 | / | **0.7259** | 0.7252 | **0.7263** | 0.7247 | 0.7259 |
| DCNN-1 | **0.2416** | **0.0372** | 0.7212 | **0.7700** | 0.7202 | **0.7438** | **0.7426** |
| DCNN-2 | **0.2358** | **0.0081** | **0.7561** | **0.7681** | **0.7576** | **0.7613** | **0.7608** |
| EfficientNetB0 | 0.3380 | / | 0.6620 | 0.6691 | 0.6570 | 0.6597 | 0.6620 |
| DCNN-1 | **0.3136** | **0.0232** | **0.6632** | **0.6880** | **0.6612** | **0.6731** | **0.6765** |
| DCNN-2 | **0.3148** | **0.0093** | **0.6759** | **0.6882** | **0.6741** | **0.6793** | **0.6813** |
| ShuffleNetV2 | 0.3763 | / | 0.6237 | 0.6267 | 0.6173 | 0.6177 | 0.6237 |
| DCNN-1 | **0.3136** | **0.0616** | **0.6248** | **0.6857** | **0.6273** | **0.6543** | **0.6602** |
| DCNN-2 | **0.3252** | **0.0023** | **0.6725** | **0.6704** | **0.6687** | **0.6689** | **0.6738** |

## 5  Discussion

**i) The parameter $\alpha$.** We know that the imprecise training samples are labeled by two judgment criteria already introduced earlier. The network extracts the meta-category features from these imprecise samples and then classifies some samples in the test set as imprecise ones. Thus, it is clear that $\alpha$ also controls the number of imprecise samples and the imprecision rate in the test set. For example, Table 3 shows the results of GoogLeNet-based DCNN on Imagewoof-5 dataset as $\alpha$ increases. We can find that $R_e$ tends to decrease as $\alpha$ increases while $R_i$ gradually increases.

Table 3: The results of GoogLeNet-based DCNN on Imagewoof-5 dataset

| Models | $\alpha$ | $R_e$ | $R_i$ | $R_a$ | $EP$ | $ER$ | $EF1$ | $B_T$ |
|---|---|---|---|---|---|---|---|---|
| DCNN-1 | 0 | 0.2221 | 0.0912 | 0.6867 | 0.7774 | 0.6877 | 0.7272 | 0.7391 |
| | 0.1 | 0.1794 | 0.1358 | 0.6848 | 0.8351 | 0.6854 | 0.7499 | 0.7628 |
| | 0.2 | 0.1862 | 0.1387 | 0.6751 | 0.8260 | 0.6773 | 0.7430 | 0.7547 |
| | 0.3 | 0.1688 | 0.1532 | 0.6780 | 0.8580 | 0.6803 | 0.7518 | 0.7660 |
| DCNN-2 | 0 | 0.2173 | 0.0087 | 0.7740 | 0.7867 | 0.7753 | 0.7791 | 0.7790 |
| | 0.1 | 0.2027 | 0.0242 | 0.7731 | 0.7944 | 0.7739 | 0.7831 | 0.7870 |
| | 0.2 | 0.2056 | 0.0223 | 0.7721 | 0.7946 | 0.7738 | 0.7817 | 0.7849 |
| | 0.3 | 0.2037 | 0.0320 | 0.7643 | 0.7974 | 0.7651 | 0.7794 | 0.7827 |

**ii) DCNN *vs.* other TBF-based networks.** As mentioned, a few TBF-based networks have been proposed. In literature (George and Sankaran [2019]), the convolutional neural networks (CNN) are used to extract sample features to transform the problem into a traditional machine learning problem, and then use TBF-based evidence $K$-NN (Denœux [2000, 2008, 2019b]) to classify the test set. In literature (Tong et al. [2021]), the CNN is also used to extract features and then these features are converted into mass functions and aggregated by Dempster's rule in a DS layer[4]. Finally, an expected utility layer performs set-valued classification based on mass functions. In fact, these methods are a hardwired combination of TBF and CNN, with the CNN essentially being used as a black box for feature extraction. Although they can characterize the imprecision in the test set, they cannot characterize the imprecision in the training set and then use this imprecision to improve the classification performance. For example, Table 4 shows the results of literature (Tong et al. [2021]) on Imagewoof-5, where parameter $\beta$ controls $R_i$ similar to $\alpha$ in DCNN. Comparing with Table 3, we can see that DCNN performs better. For DCNN, $R_i$ is much smaller and more accurate than literature (Tong et al. [2021]) when $R_e$ is about the same. Besides, other indexes are also better.

Table 4: The results of literature (Tong et al. [2021]) on Imagewoof-5 dataset

| $\beta$ | $R_e$ | $R_i$ | $R_a$ | $EP$ | $ER$ | $EF1$ | $B_T$ |
|---|---|---|---|---|---|---|---|
| 0.5 | 0.2609 | 0 | 0.7391 | 0.7684 | 0.6953 | 0.7289 | 0.7391 |
| 0.6 | 0.2367 | 0.0660 | 0.6973 | 0.7915 | 0.6967 | 0.7397 | 0.7346 |
| 0.7 | 0.2047 | 0.1406 | 0.6547 | 0.8136 | 0.7058 | 0.7544 | 0.7334 |
| 0.8 | 0.1746 | 0.2211 | 0.6043 | 0.8512 | 0.7122 | 0.7733 | 0.7235 |

**iii) Problems with this work.** **First,** the selection of parameter $\alpha$ still has not found an adaptive method; **Second,** manual screening of imprecise training samples is an inefficient method and it needs to be improved; **Third,** since there are few training samples for meta-categories, this may raise the problem of imbalanced data or be considered as a kind of missing data problem for meta-category (He and Garcia [2009]). We will gradually address these problems in our future work.

**iv) Potential research directions.** To our knowledge, this is a heuristic work based on TBF to characterize the imprecision caused by data uncertainty in the training and test sets. We then use this imprecision to constrain the network thereby improving the classification performance. The results demonstrate the feasibility of this attempt. This also leads to many potential research directions. **First,** it may be very interesting to characterize the imprecision caused by uncertainty as a branch of research in deep learning techniques; **Second,** how to improve the performance of deep neural networks by making full use of these mined precise and imprecise prior information is an open question; **Third,** how to quantify, manage, reduce and evaluate such imprecision caused by uncertainty may also become the focus of future research.

# 6 Conclusion

In this work, we presented a deep credal neural network (DCNN) for the characterization of imprecision caused by data uncertainty in the training and test sets. The proposed DCNN can exploit the imprecision in the training set to constrain the network and improve the classification performance, and its effectiveness is verified on two different image classification datasets. Afterward, we discussed some issues related to this work. In particular, since this work is heuristic, we also discussed some potential follow-up research directions.

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
