# OpenReview forum: "Deep Credal Neural Network: Characterization of Imprecision Between Categories"
_NeurIPS.cc/2021/Track/Datasets_and_Benchmarks/Round1 — Submitted to NeurIPS 2021 Datasets and Benchmarks Track (Round 1)_

### Official Review · Reviewer_n8rY · 2021-07-03
**Review for "Deep Credal Neural Networks"**

**Rating:** 4
**Confidence:** 4

**Strengths:**

- The authors add their training paradigm to various different base network architectures and compare them against each other
- An ablation study with respect to the important hyperparameter $\alpha$ is performed.
- The authors briefly discuss some limitations of their approach.

**Weaknesses:**

- The coverage of related work is not sufficient. This makes it difficult to judge the novelty and the impact of the proposed system.
  For further details see my comments in the Section 'Relation To Prior Work'

- The authors compare their approach to only a single other method, which is also based on the theory of belief functions. Since there exist numerous methods that deal with identifying label noise and ambiguities in datasets, some of them should be included in the benchmark as well to give a more complete overview of the current state of the art.

- The authors evaluate their approach on two very similar image classification datasets. Both consider fine-grained image classification with 5 different classes. Since the proposed approach is very generic and can be applied to arbitrary image classification settings, it would be great to include a more comprehensive study on more diverse image classification benchmarks.

- In general, I am not sure how well this work fits in a "Dataset and Benchmark" track. The authors do not introduce any new dataset and do not perform a comprehensive benchmark in which they compare several methods. Having read through the intended scope of this track, the closest connection that I can draw is related to the cleaning of datasets - since the proposed method enables the identification of mislabeled or ambiguous image samples. However, the experiments focus on the classification performance of their method, rather than how well ambiguous / mislabeled samples can be identified.



**Additional Feedback:**

- I am not sure if DCNN is a well-chosen abbreviation for the proposed method, since it is also commonly used to generically denote "Deep Convolutional Neural Network" (DCNN).

- In the entire manuscript, the ground truth class is denoted as $c_{ture}$. I believe this should be $c_{true}$?

**Clarity:**

The examples for mislabeled / ambiguous images in Figure 1 are not very convincing and I believe better examples could be found.
Even though the penguin and the night heron take a similar pose, it is still very clear that one image shows a penguin and the other one does not. Further, the authors should add a comment on if the image captions reflect the ground truth label or not? The caption of subfigure (c) is simply "Dogs", which is exactly what is present in the image. However, I believe the authors intended to display an image with two different dog breeds for which the image label only reflects one of the two?

**Correctness:**

The authors claim that in "most cases, samples are indistinguishable between at most two categories" (line 90). This leads to the very heavy restriction that their model can only assign at most one additional label to each image. I have doubts that this assumption holds in practice, especially in the fine-grained classification scenarios that the authors focus on. In the case of an occluded object, it might well be impossible to assign an image to any of the present classes. I would like to know the key reasons for why this restriction is necessary.

**Documentation:**

- Relevant formulas that facilitate the reproduction of the experiments are given in the manuscript.
- It says that a batch size of 32 is used and that it is reduced if the GPU runs out of memory (l. 195). Having the exact batch sizes for each model would be of interest.
- Please report the hyperparameters used for the early stopping.
- In line 214 it says that each model was evaluated 3-5 times. I suggest providing the exact number of model runs and the corresponding error bars.

**Ethics:**

- I do not have any concerns.

**Relation To Prior Work:**

The coverage of related work in the manuscript is insufficient. The authors discuss only one other approach, which is also based on the theory of belief functions. There exist numerous papers that deal with the problem of identifying label noise in image classification datasets or ambiguities in image labels, e.g.:

- https://openaccess.thecvf.com/content_cvpr_2018/papers/Lee_CleanNet_Transfer_Learning_CVPR_2018_paper.pdf
- https://openaccess.thecvf.com/content_CVPR_2019/papers/Durand_Learning_a_Deep_ConvNet_for_Multi-Label_Classification_With_Partial_Labels_CVPR_2019_paper.pdf
- https://openaccess.thecvf.com/content_ICCV_2017/papers/Rupprecht_Learning_in_an_ICCV_2017_paper.pdf

Therefore I believe that the section on related work should be substantially extended.

**Summary And Contributions:**

The authors observe that often in image classification, some images
can not be uniquely classified to a single class due to ambiguities in the images.
Furthermore, label errors might exist in the training set. To identify these two
types (i.e. ambiguous or mislabeled images), the authors propose a training paradigm
which they call "Deep Credal Neural Network" (DCNN).

During the training of a standard CNN,
the authors check if a sample is assigned its correct ground-truth label with high probability.
If this is not the case, the authors call this sample "imprecise" and assign a second class
that has also received high probability from the network as an additional label to the sample.

In a second step, the authors assign new labels to all training samples.
Labels are drawn from the powerset of the original labels. This enables
to assign an image a label of more than one of the original classes.

The authors perform experiments on two small datasets
and show that their approach compares favorably against
one other recently introduced method.

---

### Official Review · Reviewer_jYKg · 2021-07-04
**Using imprecision among categories to improve classfication.**

**Rating:** 4
**Confidence:** 3
**Correctness:** No, please see the weakness section.
**Clarity:** Yes, the paper is clearly written.

**Strengths:**

+ The paper tries to address the label imperfections, which is a super-relevant problem in the current era of data-driven learning.
+ The combination of sigmoidal epoch weighing with an exponential example weighing makes sense. The initial epochs are noisy and therefore, the initial training behaves like usual BCE + smax. However, the labels are more pronounced, and therefore, the discounting makes sense.
+ The paper is clearly written.
+ There are discussions on the failure cases and possible extensions. I appreciate the way the authors have mentioned this in their paper. This is laudable.
+ The code is provided.

**Weaknesses:**

- The assumption that the confusion happens among two classes is flawed. E.g., Classes 1, 7, and 9 are easily confused in MNIST. The flaw becomes serious when scaled to more extensive datasets such as ImageNet.
- A closely related paper that looks into the multi-class labeling probabilities is Deep Contour [1]. The positive sharing loss of [1] tries to keep a margin between the best class and other classes.
- Using toy datasets is OK, which have ~5k training and five categories. However, the label uncertainty problem is more prevalent in more extensive and more commonly used datasets such as MNIST, CIFAR-10, CIFAR-100, and ImageNet. The paper does not show any experiments on these benchmarks.
- Ablation studies are missing. It is imperative to show how performance benefits come from the multi-stage training, sigmoidal epoch weighing and exponential example weighing. I would also like to have the results when you train the networks with standard BCE and then later finetune with exponential example weighing without sigmoidal epoch weighing. That way, we know how much improvement comes from the proposed exponential example weighing scheme.
- The experiments do not report the standard deviation as required for NeurIPS. I would suggest making a bar plot with error bars instead of Table 1.
- The paper does not talk about performance relationships with the size of the network (flops), the presence and the absence of residual connections, as well as the presence/absence of Batch Normalization.
- The equation (2) of S(t) is a heuristic. Do authors evaluate alternate weighing schemes as well? E.g., a linear interpolator between t and S(t).


Reference-
[1] Deep Contour, Shen et al, CVPR 2015.

**Additional Feedback:**

- There is a typo (probably wrong latex abbreviation) of c_{true}. It appears as c_{ture} everywhere.

**Documentation:**

NA. The paper does not introduce new datasets.

**Ethics:**

NA.  The paper does not introduce new datasets.

**Relation To Prior Work:**

The paper discusses most of the previous works. I think this is a related work which should be compared with
[1] Deep Contour, Shen et al, CVPR 2015.

**Summary And Contributions:**

The paper introduces DCNN - a deep credal neural network to characterize the label imprecision in the training and test sets. The paper presents a multi-stage training of the standard classification networks. The paper multiplies a sigmoidal epoch weighing with an exponential example weighing for the first stage of label assignment. It uses this product as the weighting factor to the traditional Binary Cross-Entropy (BCE) + softmax for training the label assignment. After completing the label assignment, the paper maps each training example to n + nC2 categories and then trains with the standard BCE + softmax. Experiments on Imagewoof-5 and Flowers suggest the effectiveness of the proposed approach.

---

### Official Review · Reviewer_Zb2f · 2021-07-04
**Important paper and idea but needs to be evaluated in the main category**

**Rating:** 3
**Confidence:** 4
**Correctness:** The benchmarking results are correct …
**Clarity:** The paper is well-written overall but…

**Strengths:**

The benchmark results are important to the broader community and the idea of characterizing the imprecision in the training and test sets and to improve classification performance is interesting.

**Weaknesses:**

As mentioned earlier, the approach itself must be validated by peer-review before the benchmark results can be accepted. The paper could be written in a better manner. There are many grammatical errors and typos. For example, "whether assign an additional potential label", "he larger δ2 is the less necessary", and so on. Also, the clarity of notations can be improved.

**Additional Feedback:**

NA

**Documentation:**

Documentation is sufficient.

**Relation To Prior Work:**

NA.

**Summary And Contributions:**

The paper presents a deep credal neural network based on the theory of belief functions to characterize the imprecision in the training and test sets and thereby improve the classification performance. While the idea is certainly useful and benchmark results are presented using many well-known network structures, the method itself must be peer-reviewed in the main track where reviewers are judging the algorithmic validity and soundness of the approach. Then, the benchmark results can be presented in a DB & benchmark track. I will encourage the authors to re-submit the work in the general track. Also, the authors should perform a grammar check to improve the quality of writing.

---

### Decision · Program_Chairs · 2021-07-26

**Decision:**

Reject

**Comment:**

The authors propose a dataset to characterize the imprecision in training vs. test sets.  The reviewers appreciated the importance of the setting and several parts of the paper, but thought that the presentation lacked precision, sufficient ablation/empirical support, and potentially should have been submitted as a methods vs. dataset paper.